# Specific Relation Attention-Guided Graph Neural Networks for Joint Entity and Relation Extraction in Chinese EMR

**Yali Pang \*** , **Xiaohui Qin \*** and **Zhichang Zhang**

College of Computer Science and Engineering, Northwest Normal University, Lanzhou 730071, China
\* Correspondence: pyl@nwnu.edu.cn (Y.P.); qinxh_qj@163.com (X.Q.)

**Abstract:** Electronic medical records (EMRs) contain a variety of valuable medical entities and their relations. The extraction of medical entities and their relations has important application value in the structuring of EMR and the development of various types of intelligent assistant medical systems, and hence is a hot issue in intelligent medicine research. In recent years, most research aims to firstly identify entities and then to recognize the relations between the entities, and often suffers from many redundant operations. Furthermore, the challenge remains of identifying overlapping relation triplets along with the entire medical entity boundary and detecting multi-type relations. In this work, we propose a Specific Relation Attention-guided Graph Neural Networks (SRAGNNs) model to jointly extract entities and their relations in Chinese EMR, which uses sentence information and attention-guided graph neural networks to perceive the features of every relation in a sentence and then to extract those relations. In addition, a specific sentence representation is constructed for every relation, and sequence labeling is performed to extract its corresponding head and tail entities. Experiments on a medical evaluation dataset and a manually labeled Chinese EMR dataset show that our model improves the performance of Chinese medical entities and relation extraction.

**Keywords:** Chinese electronic medical record; joint entity and relation extraction; attention mechanism; graph neural network





## 1. Introduction

Electronic medical record (EMR) is a critical type of clinical data that contains important patient health information records. The medical entities and the relations between them contained in EMR are the crucial foundation of various health-related applications, such as diagnosing disease [1], constructing a medical knowledge base [2,3] and answering medical questions [4], etc, demonstrating that extracting entities and relations from EMRs is an essential issue in the research of medical natural language processing (NLP).

This paper studies the joint extraction of Chinese medical entities and relations from Chinese EMRs. To jointly extract entities and the relations between them is to extract structural knowledge in the form of (*Head entity*, *Relation*, *Tail entity*) from unstructured texts. For the medical problems mentioned in clinical texts, the entities are professional concepts, such as Diseases, Symptoms, Tests, and Treatment. For Chinese EMR, joint extraction of medical entities and relations is a big challenge. Table 1 shows examples of the semantic relations of medical entities in Chinese EMR. The entities and relations overlapping Chinese medical sentences can be different cases: **Normal**, **Single Entity Overlap** (SEO), and **Entity Pair Overlap** (EPO). The first example is the EPO class, which has triplets with overlapping entity pairs (铅中毒 "plumbism", 高血压 "hypertension"). The second is the SEO class, in which the single entity (空肠弯曲杆菌 "Campylobacter jejuni") overlaps in the triplets. The last is the **Normal** class, which has no overlapping entities.

**Table 1.** Examples of the relations between medical entities.

| Sentence Type | Sentence | Entity Pairs and Relation Triplets |
|---|---|---|
| EPO | 铅中毒最主要毒性表现为高血压<br>(The primary toxicity of **lead poisoning** is **hypertension**.) | (铅中毒, 临床表现, 高血压)<br>(lead poisoning, clinical manifestations, hypertension)<br>(铅中毒, 并发症, 高血压)<br>(lead poisoning, complications, hypertension) |
| SEO | 空肠弯曲杆菌是引起急性胃肠炎的主要病因, 也是最常见**CBS**前驱感染源<br>(**Campylobacter jejuni** is the leading cause of **acute gastroenteritis** and the most common source of **CBS** precursor infection.) | (空肠弯曲杆, 病因, 急性肠胃炎)<br>(Campylobacter jejuni, Cause, acute gastroenteritis)<br>(空肠弯曲杆菌, 高危因素, GBS)<br>(Campylobacter jejuni, high-risk factors, GBS) |
| Normal | 食物中毒首选氟喹诺酮类, 对大肠杆菌感染所致腹泻有一定疗效<br>(**Fluoroquinolones** are the first choice for **food poisoning**, which has a specific effect on **diarrhea** caused by *Escherichia coli* **infection**.) | (氟喹诺酮类, 药物治疗, 食物中毒)<br>(Fluoroquinolones, drug therapy, food Poisoning)<br>(腹泻, 临床表现, 大肠杆菌感染)<br>(diarrhea, clinical manifestations, Escherichia coli infection) |

### 1.1. Pipelined Extraction Model of Entity and Relation

The traditional pipelined extraction methods [5–7] treat entity and relation extractions as two separate tasks: firstly recognizing the entities [8] and then extracting the relations between them [9]. This kind of model is flexible and straightforward, yet suffers from error propagation and ignores the correlation between the two subtasks [10]. Thus, many researchers focus on building joint neural network models to extract entities and relations simultaneously. For example, ETL-Span [11] first distinguishes the entities that may be related to the target relation and then determines the corresponding tail entity and relation for each extracted entity.

### 1.2. Joint Extraction Model of Entity and Relation

Due to the limitation of traditional pipelined extraction models, many joint methods propose the extraction of overlapping entity and relation triplets. WDec [12] uses two encoder–decoder architectures to extract entities and relations jointly. Zheng et al. [13] and Sun et al. [14] use joint entity and relation extraction methods to utilize the connection between entities and relations. These models assume that there is only one relation between entity pairs and cannot accurately extract overlapping triplets.

BERT-JEORE [15] uses a parameter sharing layer to capture common features of entities and overlapping relations, assigning entity labels to each token in a sentence and then extracting entity–relation triplets. CopyRE [16], HRL [17], and RSAN [18] proposed a relation-guided joint entity and relation extraction model. However, CopyRE and HRL ignore the fine-grain information between the relation and the words in the sentence in entity recognition. RSAN introduces a large number of noise relations in the process of entity labeling, which increases the running time of the model. At the same time, these models ignore the correlation or dependence between the various relations in the sentence. Furthermore, for Chinese medical datasets, these models cannot solve the problems of medical entity boundary recognition errors and cannot make full use of medical knowledge.

### 1.3. Graph Neural Networks Based extraction Model

In recent years, graph neural networks (GNNs) [19] have been widely used in the research of entity and entity relationships of news text. Later Li et al. [20] replaced the Almeida-Pineda algorithm with a more general backpropagation and proved its effectiveness. Subsequent efforts have improved computational efficiency through local spectral convolution techniques [21,22]. Zhu et al. [23] proposed a graph neural network with generation parameters (GP-GNNs) used in relation extraction. Veličković et al. [24] presented the Graph Attention Networks (GATs), which sums neighborhood states using self-attention

layers. Zhang et al. [25] apply GATs to recognize relations between entities and directly uses the complete dependency tree as the model's input, which improves the performance of relation extraction. GraphRel [26] is proposed as a two-stage joint model based on a graph convolutional network (GCN).

In this work, we propose a Specific Relation Attention-guided Graph Neural Networks (SRAGNNs) model to jointly extract medical entities and relations in Chinese EMRs. Concerning the inner connection between different relations, we use the attention-guided GNNs to learn the correlation, and utilize multi-head attention to capture interactions between different relation types. For relation extraction, we regard it as a multi-label classification task. We believe that the relation extraction in a sentence should consider the influence between each other. Thus, the GNNs in our model are guided by attention in order to learn the correlations and dependencies between relations. By using the same multi-head attention to capture the interaction between different relation types, the model can understand the relevant semantics of each relation. Moreover, combined with the sentence context features captured by the transformer from word embedding and character embedding, the correlation strengths under different relation types are calculated respectively, and the relation classifiers with interdependent sentence meanings are learned. We use feature vectors based on specific relations to construct a particular sentence representation under each relation, and then perform sequence labeling to extract entities corresponding to specific relations.

To output the labels of medical entities and the relations between them in Chinese EMRs, we adopt the entity labeling scheme proposed by RSAN to simplify the entity labeling process. We merge the head and tail entities {H, T} in the triplet into a typical BIES sign (*begin*, *inside*, *end*, or *single*) as our entity label. We will generate separate tag sequences for sentences with multiple triplets based on different relations. In the tag sequence of a specific relation, only the corresponding head and tail entities will be labeled, and the remaining characters will be assigned the tag "O".

Our contributions to this paper can be concluded as follows:

- To propose a Specific Relation Attention-guided Graph Neural Networks (SRAGNNs) model for the joint extraction of Chinese medical entities and relations. This model captures the fine-grain semantic features between relation and Chinese word characters and can extract overlapping triplets. The model uses relations to guide entity recognition.
- The model learns the correlation between various relations through attention-guided GNNs. At the same time, by combining the sentence context features encoded by the transformer, it fully captures the relevant semantics and transforms the relation extraction into a multi-label classification task.
- For the Chinese medical datasets, we use the encoding strategy of Chinese character embedding combined with word embedding to obtain the latest results on a medical evaluation dataset and a manually labeled Chinese EMR dataset, which proved to be effective.

## 2. Methodology

### 2.1. Task Description and Overview of Method

The entity and entity relation extraction task from a Chinese EMR is to label all medical entity mentions $E = \{e_1, e_2, \ldots, e_n\}$ and classify their types $\{C(e_1), C(e_2), \ldots, C(e_n)\}$ in the EMR text, where $e_i (1 \leq i \leq n)$ is in the form of character subsequence of a sentence and $C(e_i) \in \{Disease, Symptom\ and\ sign, Test, Abnormal\ test\ result, Treatment, Drug\}$, and recognize all entity relations $\Pi = \{\pi = (h, r, t) | h, t \in E, r \in R\}$ between the entities given relation class set $R$, the definition of which is different in different datasets.

The architecture of our proposed model is shown in Figure 1. We use the Transformer [27] to encode the input sentence and extract the sentence context features. The GAT is used to learn the correlation between each relation, combined with the sentence context features encoded by the transformer, in order to fully capture the semantic information, and to extract the relation. Then, we use the feature vector based on the specific relation to

construct the particular sentence representation under each relation, and perform sequence labeling to extract the entity corresponding to the particular relation. Given a Chinese medical sentence $S = \{c_1, c_2, \cdots, c_n\}$, and a predefined relation set $R$, the task of joint entity and relation extraction is to extract relation triplets $\{\pi = (h, r, t) | h, t \in E, r \in R\}$ from $S$. The extracted triplets may share the same entity or relation, that is, the problem of overlapping triplets, where $E$ and $R$ represent the medical entity set and entity relation set, respectively, and $h$, $t$ represent the head entity and the tail entity, respectively.

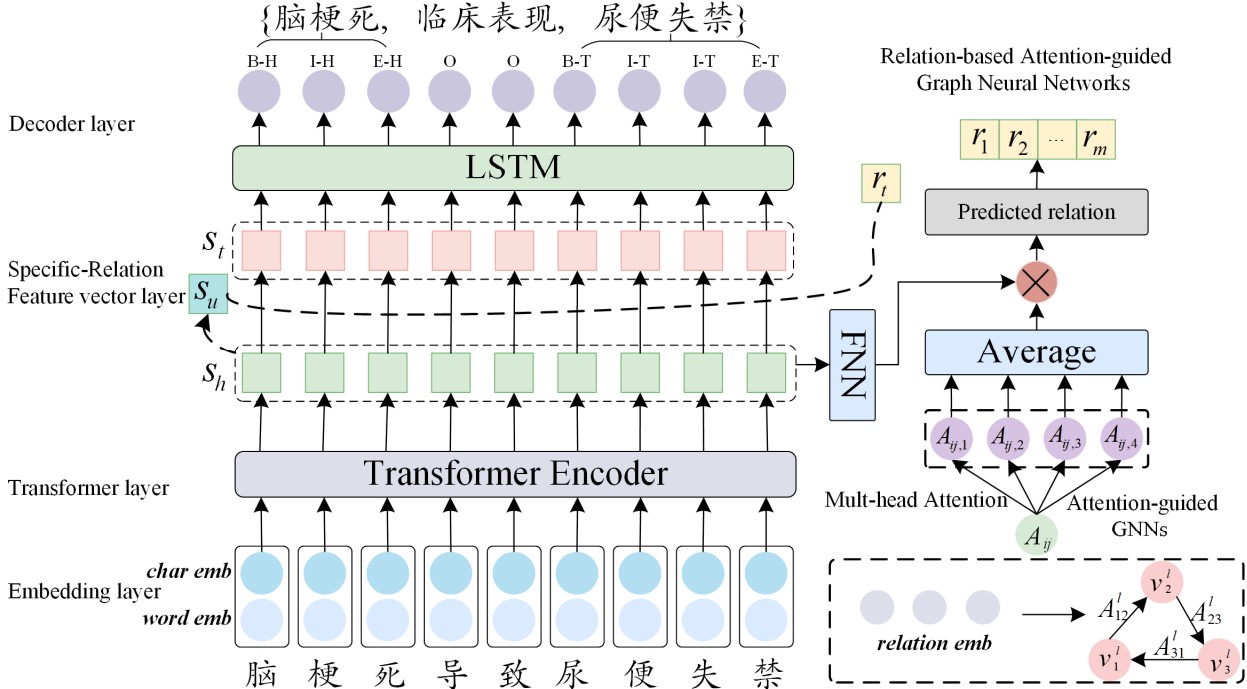

**Figure 1.** The overall architecture of the proposed SRAGNNs model. The Transformer is used to encode the input sentence and extract the sentence context features. The GAT is used to learn the correlation between each relation, combined with the sentence context features encoded by the transformer, to fully capture the semantic information, and to extract the relation. For the example input sentence in the figure, the model extracts the head entity "梗死 (cerebral infarction)" and the tail entity "尿便失禁 (urine fecal incontinence)", and combine this entity pair with $r_t$ to return the triplets {"脑梗死 (cerebral infarction)","临床表现 (clinical manifestations)","尿便失禁 (urine fecal incontinence)"}.

### 2.2. Word-Character Embedding Layer

In the model of this paper, in order to better recognize the boundary information of medical vocabulary and to avoid problems such as word segmentation errors, the encoding method of character embedding combined with word embedding is adopted. We train 100-dimensional character embeddings on 800,000 unlabeled Chinese medical EMR data, and train 100-dimensional word embeddings on a large Chinese EMR dataset and the Medical English Vocabulary Learning Manual (Second edition).

Given a Chinese medical sentence $S = \{c_1, c_2, \cdots, c_n\}$, where $c_i$ is the $i$-th character, we use character embedding to represent each character $c_i$: $x_i^c = e^c(c_i)$, where $x_i^c \in R^{d_c}$, and $e^c$ denotes the character embedding matrix. In a sentence $S$, if the set of various possible words containing the character $c_i$ is named as $ws_i$, that is $ws_i = \{w_{b_1,e_1}, w_{b_2,e_2}, \cdots, w_{b_k,e_k}\}$, where $b_j \leq i$ and $e_j \geq i$ ($j = 1, \cdots, k$), and $w_{b_j,e_j}$ is a character subsequence of $S$ which begins with $b_j$-th character and ends with $e_j$-th character, we select the longest word in $ws_i$ and use $w_i$ to represent it, that is $w_i = \text{MaxLength}(ws_i)$. Then the sentence is expressed as $rs = \{(c_1, w_1), (c_2, w_2), \cdots, (c_n, w_n)\}$. These words match the words in lexicon **D**. Lexicon **D** is pre-trained word embedding. We use word embedding to represent each

word $w_i$: $x_i^w = e^w(w_i)$, where $x_i^w \in R^{d_w}$, and $e^w$ represents the word embedding matrix. We use character embedding and its corresponding word embedding splicing as the input of the model: $x_i = [x_i^c; x_i^w]$, $x_i \in R^d (d = d_c + d_w)$.

### *2.3. Transformer Layer*

After obtaining the word-character embedding, we use the transformer to capture the features of the character and word context. The multi-head attention mechanism in the transformer can well perceive the features of the sentence context and learn the internal structure of the sentence. Its positional encoding can identify the position of different tokens, thereby capturing the sequential characteristics of the language. The word-character embedding sequence $\{x_1, x_2, \cdots, x_n\}$ is used as the input of the transformer network. The attention score is calculated using the equations given in [27]. After that, the scaled dot–product attention can be calculated. Then, we use $S_h = \{ct_1, ct_2, \cdots, ct_n\}$ to represent the context sentence features after transformer encoding, where $ct_i \in R^d$.

### *2.4. Relation-Based Attention-Guided Graph Neural Networks Layer*

The graph neural network (GNN) guided by attention learns the correlations and dependencies between relations. At the same time, combined with the sentence context features captured by the transformer from word embedding and character embedding, the relation classifiers with interdependent sentence meanings are learned.

#### 2.4.1. Graph Representation of Relation

Firstly, a fully connected graph G composed of predefined medical entity relation features in Chinese EMR is built. The graph $G$ is composed of feature description $V \in R^{m \times d_r}$ and the corresponding adjacency matrix $A \in R^{m \times m}$, where $m$ represents the number of predefined medical entity relations and $d_r$ denotes the number of dimensions. The GAT network takes as input the characteristics of the data nodes and the adjacency matrix representing the graph. The adjacency matrix is trained, in the hope that the model can determine the graph, so as to obtain the correlation between the various relations in the sentence. We model the correlation between the relations as a weighted graph, and we use GAT network learning so that together the adjacency matrix and the attention weight represent the correlation. In the context of this model, the relational embeddings of each relation in the predefined relational set act as node features, and the adjacency matrix is a learnable parameter.

#### 2.4.2. Adjacency Matrix Updating Mechanism in GNNs

In the GNNs, the adjacency matrix can be updated by different types of node update mechanisms. Then the adjacency matrix $A$ of the $l$-th layer is updated as follows.

$$A_{ij}^l = f\left(v_i^l W^l, v_j^l W^l\right) \tag{1}$$

where $f(\cdot)$ denotes an activation function (in fact, it is the ReLU activation function), $W^l$ is the weight vector of the $l$-th layer, and $v_i$ represents the relational embedding of the $i$-th predefined relation.

Each node in the graph structure represents a predefined relation. The GNNs combine all relation features with the same weights, and then the result is passed through one activation function to produce the updated node feature output. However, the influence of neighbors can vary greatly, and the attention mechanism can obtain relation importance in the correlation graph by considering the importance of neighbor relations.

When using GNNs in the model, we use multi-head attention mechanism, which uses $K$ different heads to describe the information between relations. The operation of this layer is independently copied $K$ times (each copy is completed with different parameters), and finally the average value of $K$ attention is taken. Among them, the initial embedding of the

node (that is, the representation of the 0th layer) is the vector that encodes the feature of the node.

$$v_i^{l+1} = \sigma \left( \frac{1}{k} \sum_{k=1}^{K} \sum_{j=1}^{m} A_{ij,k}^l v_i^l w^l \right) \tag{2}$$

The output from the last layer is the attended relation features $V_{gat} = R^{m' \times d_k}$, where $m'$ is the predicted number of potential relations for each sentence. The sentence embedding encoded by the transformer passes through the normalization layer and is denoted as $F$. Then, we multiply the sentence feature embedding with the relation feature to obtain the final prediction score, and perform the relation classification as follows:

$$y^r = F \odot V_{gat} \tag{3}$$

After extracting the relations, we denote entities relations existing in each sentence as $R' = \{r_1, r_2, \cdots, r_{m'}\}$, where $r_t \in R^{d_r}$ is the trainable embedding of the $t$-th relation.

### 2.5. Specific Relation Feature Vector Generation Layer

The words and characters in a sentence have different semantics in different relations. To this end, we assign weights to the contexts under each relation. But most of the words–characters in the sentence are useless for entity recognition under a specific relation. In this case, by using the attention mechanism, it is easy to make the context weight distribution under each relation flat, therefore we have to sharpen its weight value. We introduce a sharpening coefficient $\lambda$ to solve this problem. The smaller the $\lambda$, the sharper the weight, and the more obvious the weight distribution of the context in the sentence.

$$\alpha_i^t = f(w_r r_t + w_u S_u + w_h h_i) \tag{4}$$

$$e_i^t = \frac{\exp\left(\frac{\alpha_i^t}{\lambda}\right)}{\sum_{j=1}^{n} \exp\left(\frac{\alpha_j^t}{\lambda}\right)} h_i \tag{5}$$

where $w_r = R^{d \times d_r}$, $w_u, w_h = R^{d \times d}$ are trainable parameters. Here $S_u$ indicates the global representation of the sentence: $S_u = \frac{1}{n} \sum_{i=1}^{n} h_i$. In this way, we have assigned relational information and sentence global information for each token. It can measure the importance of each character to the expression of the relation, which is conducive to subsequent entity recognition. Therefore, the sentence under the specific relation $t$ is expressed as $S_t = \{e_1^t, e_2^t, \cdots, e_n^t\}$.

### 2.6. Entity Decoding Layer

We perform specific relation decoding on the weighted token information. Here we run the BiLSTM network on the sequence $S_t$ to decode and map each word to the label space.

$$h_i^t = [\overrightarrow{LSTM}(e_i^t); \overleftarrow{LSTM}(e_i^t)] \tag{6}$$

Conditional Random Field (CRF) is employed for the task of sequence labeling. First, $Y$ is used to denote all possible sequence tag types of sentence $S$. For the sequence tag type set $Y = \{y_1, y_2, \cdots, y_n\}$, given the sentence $s$ inputted to the CRF model, its probability is defined as:

$$P(y_i^t) = \frac{\prod_{i=1}^{n} \phi_i(y_{i-1}^t, y_i^t | s)}{\sum_{y^{t\prime} \in Y(s)} \prod_{i=1}^{n} \phi_i\left(y_{i-1}^{t\prime}, y_i^{t\prime} | s\right)} \tag{7}$$

where $\phi_i\left(y_{i-1}^t, y_i^t \mid s\right) = \exp\left(W_y e_i^t + b_{y^{t'},y^t}\right)$, $W_{y^t}$ and $b_{y^{t'},y^t}$ are trainable parameters corresponding to the label pair $\left(y^{t'}, y^t\right)$, and $y^{t'}$ is all possible label sequences. $P\left(y_i^t\right)$ represents the probability of the predicted label of the $i$-th word under the relation $r_t$.

We use negative log-likelihood loss function to train our model. We denote the ground truth labels under relation $r_t$ as $\left\{\hat{y}_1^t, \hat{y}_2^t, \cdots, \hat{y}_n^t\right\}$, then loss can be defined as:

$$L = \frac{1}{m \times n} \sum_{t=1}^{m} \sum_{i=1}^{n} -log P\left(y_i^t = \hat{y}_i^t\right) \tag{8}$$

## 3. Experiments

### 3.1. Experimental Setup

#### 3.1.1. Datasets

This paper evaluates our model on two Chinese medical datasets. Their statistics are listed in Table 2.

**Table 2.** Statistics of the datasets.

| Datasets | Relation Types | Train Sentences | Dev Sentences | Test Sentences |
|----------|----------------|-----------------|---------------|----------------|
| CEMR | 15 | 35,146 | 4222 | 4389 |
| CMeIE | 44 | 14,320 | 1790 | 1814 |

**CEMR (Chinese Electronic Medical Record) dataset**: a corpus constructed by ourselves in order to facilitate the study of Chinese EMR entity and relation extraction tasks and future work on related topics. In this dataset, the relations between medical entities are divided into fifteen categories. The normalization of the labeling process refers to a large number of annotation guidelines including the medical relation annotation specifications of 2010 i2b2/VA Challenge [28]. All EMRs are from Third-Class A-Level hospitals in Gansu Province, China, and are real electronic medical records. These data contain 4000 EMR across 14 medical departments.

**CMeIE (Chinese Medical Information Extraction) dataset**. The CMEIE dataset [29] includes pediatric training corpus and one hundred common diseases training corpus. The pediatric training corpus is derived from 518 pediatric diseases, and the one hundred common diseases training corpus is derived from 109 common diseases. We re-adjust the ratio of train dataset, development dataset, and test dataset in the original dataset.

#### 3.1.2. Experimental Settings

This paper uses the TensorFlow 2.0 framework to implement the SRAGNNs model. All experiments are carried out on a computer with a memory of 64G, a graphics card of Tesla P4, and a system of Ubuntu Server 18.04. The parameters of the SRAGNNs models are shown in Table 3.

**Table 3.** Experimental parameters.

| Parameter Description | Value |
|-----------------------|-------|
| Number of transformer encoder blocks | 6 |
| Layers of GAT | 3 |
| Dimension of character embedding | 100 |
| Dimension of word embedding | 100 |
| Dimension of relational embedding | 200 |
| Learning rate | 0.001 |
| Batch size | 32 |
| Number of epochs | 50 |
| Dropout rate | 0.1 |

3.1.3. Comparison Methods

To achieve a comprehensive and comparative analysis of our model, we compare it with a series of models:

- **CopyRe** is a seq2seq model with a copy mechanism, and uses the copy mechanism to generate triplets in a sentence in order. The model can jointly extract entities and relations from sentences of any type.
- **GraphRel** is an end-to-end relation extraction model. The model constructs a complete word graph for each sentence, using graph convolutional networks (GCNs) to jointly extract entities and relations.
- **HRL** applies a hierarchical paradigm. The entire extraction process is decomposed into a two-level reinforcement learning strategy hierarchy, which is used for relation detection and entity extraction, respectively. It first performs relation detection as a high-level reinforcement learning process, and then identifies the entity as a low-level learning process.
- **ETL-Span** decomposes the joint extraction task into two interrelated subtasks. Firstly, distinguishing all head entities, and then identifying the corresponding tail entities and relations.
- **WDec** designed a new representation scheme and uses the seq2seq model to generate the entire relation triplet.
- **RSAN** uses the relation-aware attention mechanism to construct a specific sentence representation for each relation, and then performs sequence labeling to extract its corresponding head and tail entities.
- **BERT-JEORE** uses a BERT-based parameter sharing layer to capture joint features of entities and overlapping relations, assigning entity labels to each token in a sentence, and then extracting entity-relation triplets.

The datasets used in our experiment are Chinese medical datasets. In the input part of the model, the combination of character and word information in the Chinese datasets is more complicated than that of the English datasets. The above models are all experiments done on the English datasets, so when we apply them to the Chinese medical data set, the input of these models all use Chinese character embedding.

*3.2. Results*

We adopt *Precision*, *Recall*, and *F*1 score to evaluate the performances [30]. A triplet is considered to be correctly extracted if and only if the head and tail entity and the relation type between them are exactly matched.

Table 4 shows the experimental results on the two datasets. We can find that the experimental results of our model on the two datasets are better than other baseline models. The *F*1 score of the SRAGNN model on the CEMR dataset is 1.88% better than that of the BERT-JEORE, and the *F*1 score on the CMeIE dataset is 1.41% higher than the BERT-JEORE. The *F*1 score of the SRAGNNs model is 3.91% better than the RSAN model on the CEMR dataset, and the *F*1 score is 2.96% better than the RSAN model on the CMeIE dataset. The RSAN model also uses entity relations to guide entity recognition, but the design of the RSAN model is for the English dataset in the general domain, so when it is directly applied to the Chinese medical dataset, the experimental results of entity and relation extraction will not be optimal. WDec adopts the seq2seq model to generate relation triplets and removes the duplicate triplets and fragmentary triplets via post-processing. Therefore, WDec achieves high precision on two datasets, while its recall is unsatisfactory, thus the *F*1 score of the WDec model does not improve.

**Table 4.** Results on CEMR and CMeIE.

| Model | CMeIE | | | CEMR | | |
|---|---|---|---|---|---|---|
| | *Precision* (%) | *Recall* (%) | *F1* (%) | *Precision* (%) | *Recall* (%) | *F1* (%) |
| CopyRe | 45.81 | 36.50 | 40.63 | 51.05 | 47.50 | 49.21 |
| GraphRel | 53.26 | 50.02 | 51.59 | 56.86 | 53.52 | 55.14 |
| HRL | 58.93 | 52.32 | 55.43 | 62.33 | 58.57 | 60.39 |
| ETL-Span | 66.59 | 52.50 | 58.71 | 72.93 | 55.37 | 62.95 |
| WDec | **68.05** | 56.37 | 61.66 | **77.04** | 61.04 | 68.11 |
| RSAN | 65.02 | 61.44 | 63.18 | 73.05 | 69.80 | 71.39 |
| BERT-JEORE | 66.21 | 63.31 | 64.73 | 72.53 | 74.33 | 73.42 |
| SRAGNNs(#layers = 2) | 67.58 | **64.77** | **66.14** | 76.44 | **74.35** | **75.30** |

To further explore the performance of SRAGNNs model for extracting Chinese medical entities and relations, we analyze the performance of different elements (H, R, T) in medical entity-relation triplets on the CEMR dataset, where H represents the head entity, T represents the tail entity, and R represents the entity relation. Table 5 shows the experimental results of the SRAGNNs model on triplets with different relations.

**Table 5.** Results for relational triplet elements.

| Element | *Precision* (%) | *Recall* (%) | *F1* (%) |
|---|---|---|---|
| H | 84.49 | 85.98 | 85.23 |
| T | 83.86 | 85.29 | 84.57 |
| R | 88.02 | 88.80 | 88.41 |
| (H, T) | 75.26 | 84.76 | 79.73 |
| (H, R) | 77.28 | 80.38 | 78.80 |
| (R, T) | 74.53 | 84.58 | 79.24 |
| (H, R, T) | 76.44 | 74.35 | 75.30 |

From the experimental results in Table 5, it can be seen that the performance gap of the SRAGNNs model in recognizing only the head entity and only the tail entity is basically the same as the performance gap between (H, R) and (R, T). This demonstrates the effectiveness of the SRAGNNs model in recognizing head entity and tail entity. Furthermore, it can be found that the *F*1 scores on (H, T) and (H, R, T) are quite different because the SRAGNNs model is based on relation-guided entity recognition. At the same time, the overlap of entity–relation triplets also directly leads to the correct extraction probability of entity pairs being greater than the correct extraction probability of entity–relation triplets. This also shows that in the process of entity–relation triplets extraction, most of the entity pairs are correctly recognized.

To verify the ability of the SRAGNNs model to extract relational triplets from sentences with different degrees of overlap, we conducted further experiments on the CEMR dataset. The experimental results are shown in Table 6. The performance of several models, CopyRe, GraphRel, HRL, and ETL-Span, decreases as the number of relation triplets in the sentence increases. In contrast, the performance of the SRAGNNs model is significantly improved when extracting multiple triplets. The SRAGNNs model outperforms the BERT-JEORE model by 1.99% and 1.06%, respectively, when the overlapping relation triplets in a sentence are three and four. When a sentence contains three or more overlapping relation triplets, the SRAGNNs model outperforms other baseline systems. Therefore, the SRAGNNs model is more suitable for handling complex overlapping relation triplets than the baseline models.

**Table 6.** Relation extraction on sentences with different numbers of triplets.

| Model | N = 1(%) | N = 2(%) | N = 3(%) | N = 4(%) | N ≥ 5 (%) |
|---|---|---|---|---|---|
| CopyRe | 59.37 | 49.33 | 43.62 | 46.19 | 25.49 |
| GraphRel | 64.61 | 54.83 | 49.37 | 48.31 | 32.86 |
| HRL | 70.87 | 62.21 | 56.58 | 58.93 | 39.87 |
| ETL-Span | 71.83 | 68.12 | 57.64 | 69.44 | 57.73 |
| WDec | 67.66 | 70.20 | 68.35 | 75.67 | 43.50 |
| RSAN | 67.83 | 71.54 | 74.22 | 80.92 | 62.41 |
| BERT-JEORE | 74.21 | 72.36 | 75.47 | 80.34 | 64.72 |
| SRAGNNs(#layers = 2) | **76.18** | **74.53** | **77.46** | **81.40** | **66.93** |

The advantages of the SRAGNNs model are as follows: for Chinese medical texts, the model uses a combination of pre-trained medical character and word embedding as an input form, which avoids the problem of word segmentation errors. At the same time, the addition of segmentation features can make good use of the boundary information of characters and words. The model pays attention to the correlations and dependencies between relations, while capturing and sharpening the fine-grain features in the relations and Chinese characters and words. Our model can effectively capture the overlapping triplets.

*3.3. Ablation Study*

In our model, in view of the particularity of Chinese medical data, we used pre-trained medical character embedding combined with pre-trained medical word embedding as the input of the SRAGNNs model, which improves the performance of the entire model. In order to better capture the correlation between relations, we used the attention mechanism when using GNNs. We conducted ablation experiments to prove the effectiveness of word embedding in the model and the effectiveness of the attention mechanism in relation classification. We checked the influence of the number of layers of the GNNs on the performance of the model. At the same time, the effectiveness of the relation-based attention-guided graph neural network layer in the model is verified. The experimental results are shown in Table 7.

**Table 7.** Ablation study of SAAGNNs on CEMR dataset.

| Datasets | Train Sentences | Dev Sentences | Test Sentences |
|---|---|---|---|
| **SRAGNNs(#layers = 2)** | **76.44** | **74.35** | **75.30** |
| - Word Embedding | 74.59 | 72.99 | 73.78 |
| - Attention | 69.42 | 73.20 | 71.26 |
| - RAGNN Layers [1] | 63.30 | 67.66 | 65.41 |
| SRAGNNs (#layers = 1) | 75.81 | 73.82 | 74.80 |
| SRAGNNs (#layers = 3) | 75.70 | 74.15 | 74.92 |

[1] Relation-Based Attention-Guided Graph Neural Networks Layers.

In order to verify the impact of word embedding on sentence context features, we deleted the word embedding in the input layer in the SRAGNNs model with two layers of GNNs. In order to verify the influence of the attention mechanism in Relation-Based Attention-guided Graph Neural Networks on the model, we deleted the attention mechanism in the GNNs. To verify the effectiveness of the relation-based attention-guided graph neural networks layer in the model, we removed this module from the model. When the latent relation prediction layer is removed, each relation in the predefined relation set is used to construct the specific relation feature vector. In order to see the influence of the GNNs layer number on the model performance, we set the GNNs layer number to 1, 2 and 3, respectively.

Through the experimental results, we found that when the GNNs has two layers, the model performance is better. The attention mechanism in word embedding and relation classification also has a great influence on the performance of the model. As shown in the experimental results in Table 7, without the relation-based attention-guided graph neural

networks layer, the *F*1 score of the SRAGNNs model is significantly reduced. This is due to the fact that there are no potential relations in the sentence, resulting in an increase in the number of predicted entity pairs, which will greatly increase relation redundancy. With the increase in the number of medical relations in Chinese EMR, the training and inference time also increases substantially. Through this experiment, the effectiveness of several modules in the SRAGNNs model is demonstrated, which is beneficial to the improvement of the overall experimental performance.

From Table 7, we can see that word embedding also has a certain impact on the performance of the model. To further verify the impact of word information in Chinese electronic medical records on model performance, we analyzed the correlation information between character and word in Chinese EMRs (as shown in Figure 2. Correlation between word information and character information in Chinese EMR.). It can be seen that the semantic relevance weight of each character in Chinese EMR and its corresponding related words will be higher. Moreover, in the context of EMRs, the distance between each character also affects the semantic correlation weight between the character information and each word information in the sentence. This also shows that word information is important in Chinese medical entity and relation extraction.

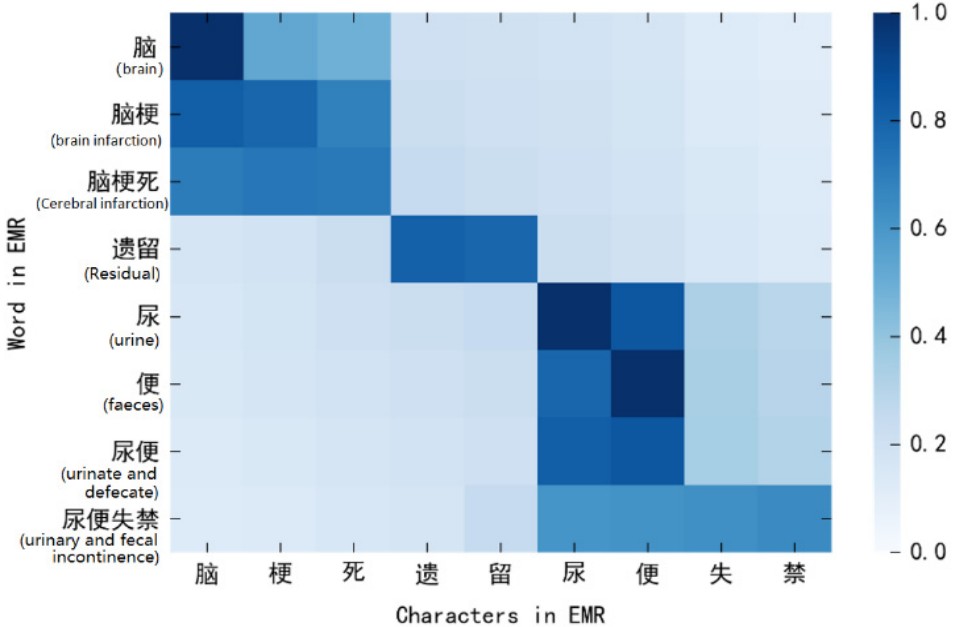

**Figure 2.** Correlation between word information and character information in Chinese EMR.

### 3.4. Analysis on Overlapping Cases

In order to verify the ability of SRAGNNs model to extract multiple triplets, we conducted further experiments on the CEMR dataset and analyzed the experimental results on different sentence types. Figure 3 shows the experimental results of each model in different sentence types. In the case of entity overlap, the performance of our proposed model outperforms all other methods. This shows that we regard relation extraction as a multi-label classification problem, and it is useful to use relations to guide entity recognition.

We can see that ETL-Span performs relatively well in Normal class. This is because its decomposition strategy is designed to be more suitable for the Normal cases. At the same time, we found that because the RSAN model is also a relation-guided joint entity and relation extraction model, the RSAN model has a better ability to extract triplets when various types of entities overlap. However, Chinese medical entities and relations are special and complex, and the performance of extracting triplets using the RSAN method is not comparable to SRANNs.

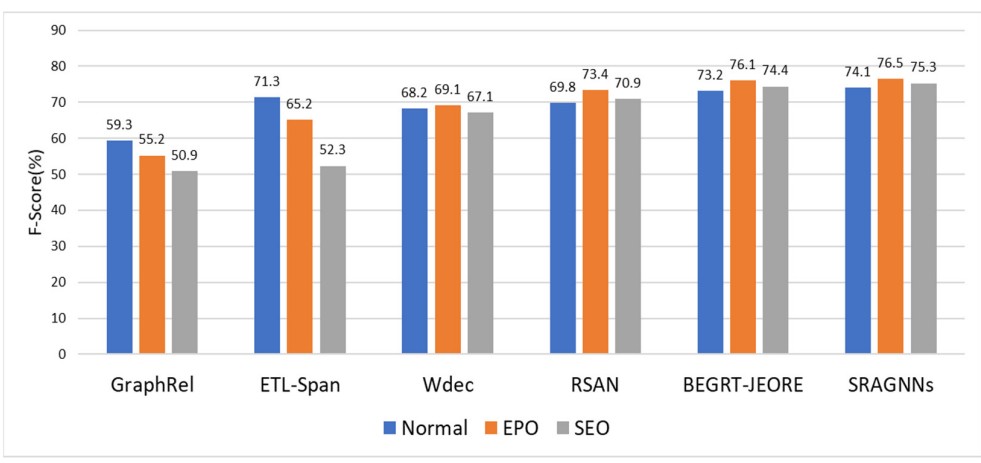

**Figure 3.** Results on different sentence types according to the degree of overlapping.

In order to verify that the SRAGNNs model can better extract entity–relation triplets, we list some examples from the CEMR dataset in Table 8. It can be seen that the SRAGNNs model can extract triplets where one or more entities in each sentence overlap. This demonstrates the effectiveness of the SRAGNNs model in addressing entity extraction and overlapping relation problems.

**Table 8.** Case study of SRAGNNs model on CEMR dataset.

| | |
|---|---|
| Sentence S1 | 脑梗死遗留尿便失禁<br>(**Urinary and fecal incontinence** after **cerebral infarction**.) |
| SRAGNNs | (脑梗死, 临床表现, 尿便失禁)<br>(Cerebral infarction, clinical manifestations, Urinary and fecal incontinence) |
| Ground truth | (脑梗死, 临床表现, 尿便失禁)<br>(Cerebral infarction, clinical manifestations, Urinary and fecal incontinence) |
| Sentence S2 | 铅中毒最主要毒性表现为高血压<br>(The main toxicity of **lead poisoning** is **hypertension**.) |
| SRAGNNs | (铅中毒, 临床表现, 高血压)<br>(lead poisoning, clinical manifestations, hypertension) |
| Ground truth | (铅中毒, 并发症, 高血压)<br>(lead poisoning, complications, hypertension)<br>(铅中毒, 临床表现, 高血压)<br>(lead poisoning, clinical manifestations, hypertension) |
| Sentence S3 | **ADV**肺炎可发展为闭塞性细支气管炎 **(B0)**, 导致反复喘息<br>(**ADV pneumonia** may progress to bronchiolitis obliterans (**B0**), leading to **repeated wheezing**.) |
| SRAGNNs | (ADV肺炎, 转化, 闭塞性细支气管炎)<br>(ADV pneumonia, transformation, bronchiolitis obliterans)<br>(闭塞性细支气管炎, 同义词, BO)<br>(Bronchiolitis obliterans, synonym, BO)<br>(ADV肺炎, 临床表现, 反复喘息)<br>(ADV pneumonia, clinical manifestations, recurrent wheezing) |
| Ground truth | (ADV肺炎, 转化, 闭塞性细支气管炎)<br>(ADV pneumonia, transformation, bronchiolitis obliterans)<br>(闭塞性细支气管炎, 同义词, BO)<br>(Bronchiolitis obliterans, synonym, BO)<br>(ADV肺炎, 临床表现, 反复喘息)<br>(ADV pneumonia, clinical manifestations, recurrent wheezing) |

In Table 8. (1) Sentence S1 is a normal class, and the entity-relation triplets are correctly recognized. (2) Sentence S2 is EPO class. The SRAGNNs model only extracted a triplet of {"铅中毒 (lead poisoning)", "临床表现 (clinical manifestations)", "高血压 (hypertension)"}, and the relation of "并发症 (complications)" was not extracted. A description such as "main toxicity" in the sentence leads directly to this relation of clinical presentation. At the same time, the relation "complication" is not directly mentioned in the context of the sentence. Therefore, the SRAGNNs model based on the semantic information of the sentence excludes this complication relation triplet, alleviating the noise problem of remote supervision. This requires the SRAGNNs model to further learn the deep semantic information in this paper. (3) The third sentence is SEO category. In this example, three triplets are involved. Among them, "ADV肺炎 (ADV pneumonia)" and "闭塞性细支气管炎 (bronchiolitis obliterans)" are overlapping entities. The SRAGNNs model identifies these triplets through overlapping relation extraction.

## 4. Conclusions

In this work, we propose a joint extraction model of entities and the relations among them, called SRAGNNs, which aims to identify overlapping triplets. We use both pre-trained medical Chinese character and word embedding as input to make extensive use of the information entailed in Chinese characters and words in order to avoid the problem of word segmentation error. Using the attention-guided graph neural networks to learn the correlation between each relation, combined with the context features captured by the transformer, it learns the relation classifier of interdependence between sentence meanings. At the same time, feature vectors based on specific relations are used to construct particular sentence representations, and CRF is used to extract entity pairs under particular relations. Experimental results on two medical datasets show that our model can improve the performance of medical entity and relation extraction from Chinese EMRs.

**Author Contributions:** Conceptualization, Y.P.; methodology, X.Q.; software, X.Q.; validation, Y.P.; formal analysis, Z.Z.; investigation, Z.Z.; resources, Z.Z.; data curation, X.Q.; writing original draft preparation, X.Q.; writing review and editing, Y.P.; visualization, X.Q.; supervision, Z.Z.; project administration, Y.P. All authors have read and agreed to the published version of the manuscript.

**Funding:** This research was funded by the National Natural Science Foundation of China (NO. 62163033), the Natural Science Foundation of Gansu Province, China (NO. 21JR7RA781, NO. 21JR7RA116), Lanzhou Talent Innovation and Entrepreneurship Project, China (NO. 2021-RC-49) and Northwest Normal University Major Research Project Incubation Program, China (No. NWNU-LKZD2021-06).

**Institutional Review Board Statement:** Not applicable.

**Informed Consent Statement:** Not applicable.

**Data Availability Statement:** Not applicable.

**Conflicts of Interest:** The authors declare no conflict of interest.

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
