# Peer review of "Specific Relation Attention-Guided Graph Neural Networks for Joint Entity and Relation Extraction in Chinese EMR"

_applsci, doi:10.3390/app12178493_

Round 1
Author Response
Dear reviewer,
Thank you very much for your careful and solid review and all comments.
We have carefully revised the paper. Specific modifications are as follows:
â–ª The first paragraph of the introduction needs syntactic correcting!
The syntactic expression of the first paragraph is modified.
â–ª Long introduction and overlapped with related works
The related works of the manuscipt is merged into the introduction.
â–ª Table 1 and also when introduced in text: the definition of 3 types of relations is not cited.
The original expression is not accurate enough. So the sentence was modified.
â–ª Figure1: insufficient architecture explanation
The explanation of Figure 1 is supplemented.
â–ª In Methodology: problem definition is not completely clear but understandable! I suggest adding a subsection for problem definition and elaborating this with more formal expressions and accurate descriptions.
The formal description of the problem is added.
â–ª References should be cited at the end of the sentence (line 55).
Modified.
â–ª Line 175: the sentence does not have a reference.
Modified.
â–ª Line 184: the ws_i is described first and then shown in the formulas while we expect vice versa.
Modified.
â–ª Formula 1,2,3,4 are not newly introduced, better to be referred to the original paper rather than providing four formulas in a row.
Original Formula 1, 2, 3, 4 are deleted and referred to the original paper.
â–ª Formula 4, w_o is not defiened!
The formula was deleted, since transformer model and its formulas have been described in the paper that proposed this model. The description and the related formulas about this architecture are referred in this manuscript.
â–ª Line 208: the i in h_i references to what? We have h, but we don’t have h_i
It should be c_i. Modified.

Reviewer 2 Report
The approach is interesting, and the research very promising. The authors propose a SRAGNN model to improve the performance of Chinese medical entities and the extraction of their relations. The model is applied on two medical datasets. Through experimental results, the model is calibrated and its suggestions is improved.
It would be great that the paper also discusses how the research and proposed model could be scaled outside the Chinese context, and also what are the limitations and future developments of this research/model.
Author Response
Dear reviewer,
Thank you very much for your comments. We have made some modifications to the paper. In addition, we will also validate our model on other datasets.

Reviewer 3 Report
Comments to Authors
The manuscript entitled “Specific Relation Attention-guided Graph Neural Networks for Joint Entity and Relation Extraction in Chinese EMR” by Yali Pang et al., propose a Specific Relation Attention-guided Graph Neural Networks (SRAGNNs) model to extract entities and their relations in Chinese EMR; providing a broad explanation of the model, evaluating it with two Chinese medical data-sets, and then comparing the results obtained with 7 other methods.
In my view, the article presents a good development and analysis of the results obtained. The manuscript is interesting and provides a good model that would seem to improve the performance of Chinese medical entity and relation extraction. So, it would be very useful for mathematic, informatic and medical researcher, and for the scientific community in general. Therefore, it could be Accept with Minor Revision. Some Minor Revision are described below.
MINOR ISSUES:
- Although the introduction is very good, both for being current and complete, I think that its structure could be improved to avoid repetitions in the text (for example lines 48 to 53 and 116 to 121). For me, it is not necessary to divide the introduction into a new section called "Related Work", since the information given in this new section is still an introduction: Description of the methodologies/previous works and their relationship with the author’s work. From my point of view, points 2.1, 2.2 and 2.3 could be incorporated in the introduction as 1.1, 1.2 and 1.3 before line 83.
- The caption of Figure 1 should include a more complete description, so that it is easier to interpret without having to search the text.
- In the graph of Figure 3, the numbers should be arranged so that they do not overlap.
- Some comparison with the other 7 methods could be added to the conclusions, since it is only mentioned that the proposed model is used on two medical datasets.
Author Response
Dear reviewer,
Thank you very much for your solid and carefully comments. We have carefully revised the paper. Specific modifications are as follows:
- Although the introduction is very good, both for being current and complete, I think that its structure could be improved to avoid repetitions in the text (for example lines 48 to 53 and 116 to 121). For me, it is not necessary to divide the introduction into a new section called "Related Work", since the information given in this new section is still an introduction: Description of the methodologies/previous works and their relationship with the author’s work. From my point of view, points 2.1, 2.2 and 2.3 could be incorporated in the introduction as 1.1, 1.2 and 1.3 before line 83.
Thank you for your valuable opinion. We incorporated the related work part into the introdunction.
- The caption of Figure 1 should include a more complete description, so that it is easier to interpret without having to search the text.
The description of Figure 1 is supplumented.
- In the graph of Figure 3, the numbers should be arranged so that they do not overlap.
Figure 3 is modified.
- Some comparison with the other 7 methods could be added to the conclusions, since it is only mentioned that the proposed model is used on two medical datasets.
Terrible Sorry, we don't quite understand the comment. In order to make the performance comparison more intuitive, the results of different methods are still placed in the same table.
